# R3M: A Universal Visual Representation
# for Robot Manipulation

**Suraj Nair**[1,*], **Aravind Rajeswaran**[2], **Vikash Kumar**[2], **Chelsea Finn**[1], **Abhinav Gupta**[2]

[1]Stanford University, [2]Meta AI

**Abstract:** We study how visual representations pre-trained on diverse human video data can enable data-efficient learning of downstream robotic manipulation tasks. Concretely, we pre-train a visual representation using the Ego4D human video dataset using a combination of time-contrastive learning, video-language alignment, and an L1 penalty to encourage sparse and compact representations. The resulting representation, R3M, can be used as a frozen perception module for downstream policy learning. Across a suite of 12 simulated robot manipulation tasks, we find that R3M improves task success by over 20% compared to training from scratch and by over 10% compared to state-of-the-art visual representations like CLIP and MoCo. Furthermore, R3M enables a Franka Emika Panda arm to learn a range of manipulation tasks in a real, cluttered apartment given just 20 demonstrations.

**Keywords:** Visual Representation Learning, Robotic Manipulation

## 1   Introduction

How do we train a robot to complete a manipulation task from images? A standard and widely used approach is to train an end-to-end model from scratch using data from the same domain [1]. However, this can be prohibitively data intensive and severely limits generalization. In contrast, computer vision and natural language processing (NLP) have recently taken a major departure from this "tabula rasa" paradigm. These fields have focused on using diverse, large-scale datasets to build *reusable, pre-trained representations*. Such models have become ubiquitous; for example, visual representations from ImageNet [2] can be reused for tasks like cancer detection [3], and pre-trained language embeddings like BERT [4] have been used for everything from medical coding [5] to visual question answering [6]. Such an equivalent of an ImageNet [2] or BERT [4] model for robotics, that can be readily downloaded and used for any downstream simulation or real-world manipulation task, has remained elusive.

Why have we struggled in building this universal representation for robotics? Our conjecture is that we haven't converged on using the appropriate datasets for robotics. Collecting large and diverse datasets of robots interacting with the physical world can be costly, even without human annotation. Recent attempts at creating such datasets [7, 8, 9, 10], consist of a limited number of tasks in at most a handful of different environments. This lack of diversity and scale makes it difficult to learn representations that are broadly applicable. At the same time, the recent history of computer vision and NLP suggests an alternate route for robotics. The best representations in these fields did not arise out of task-specific and carefully curated datasets, but rather the use of abundant in-the-wild data [4, 11, 12, 13]. Analogously, for robotics and motor control, we have access to videos of humans interacting in semantically interesting ways with their environments [14, 15, 16]. This data is large and diverse, spanning scenes across the globe, and tasks ranging from folding clothes to cooking a meal. While the embodiment present in this data differs from most robots, prior work [17, 18] has found that such human video data can still be useful for learning reward functions. Furthermore, domain gap has not been a major barrier for using pre-trained representations in traditional vision and NLP tasks. In this backdrop, we ask the pertinent question: *can visual representations pre-trained on diverse human videos enable efficient downstream learning of robotic manipulation skills?*

6th Conference on Robot Learning (CoRL 2022), Auckland, New Zealand.

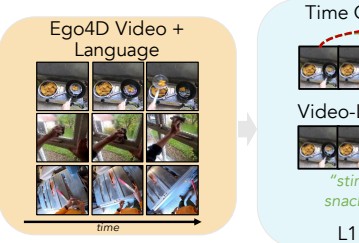
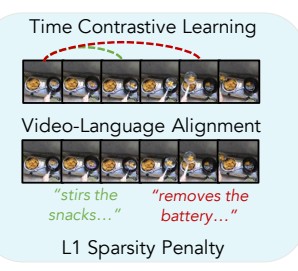
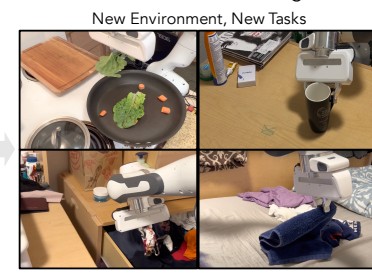

Figure 1: **Pre-Training Reusable Representations for Robot Manipulation (R3M)**: We pre-train a visual representation using diverse human video datasets like Ego4D [16], and study its effectiveness for downstream robot manipulation tasks. Our representation model, R3M, is trained using a combination of time-contrastive learning, video-language alignment, and an L1 sparsity penalty. We find that R3M enables data efficient imitation learning across several simulated and real-world robot manipulation tasks.

We hypothesize that a good representation for vision-based robotic manipulation consists of three components. First, it should contain information necessary for physical interaction, and thus should capture the temporal dynamics of the scene (i.e. how states might transition to other states). Second, it should have a prior over semantic relevance, and should focus on task relevant features like objects and their relationships. Finally, it should be compact, and not include features irrelevant to the above criteria (e.g. backgrounds). Towards satisfying these three criteria, we study a representation learning approach that combines (1) time contrastive learning [19] to learn a representation that captures temporal dynamics, (2) video-language alignment to capture semantically relevant features of the scene, and (3) L1 and L2 penalties to encourage sparsity. Our experimental evaluation in Section 4.4 finds that all three components are important for training highly performant representations.

In this work we empirically demonstrate that representations pre-trained on diverse human video datasets like Ego4D [16] can enable efficient downstream policy learning for robotic manipulation. Our core contribution is an artifact – the pre-trained vision model – that can be used readily in other work. Concretely, we pre-train a **r**eusable **r**epresentation for **r**obotic **m**anipulation (R3M), which can be used as a frozen perception module for downstream policy learning in simulated and real robot manipulation tasks. We demonstrate this via extensive experimental results across three existing benchmark simulation environments (Adroit [20], Franka-Kitchen [21], and MetaWorld [22]) as well as real robot experiments in a cluttered apartment setting. R3M features outperform a wide range of visual representations like CLIP [12], (supervised) ImageNet [2], MoCo [23, 24], and learning from scratch by over 10% when evaluated across 12 tasks, 9 viewpoints, and 3 different simulation environments. On a Franka Emika Panda robot, R3M enables learning challenging tasks like putting lettuce in a pan and folding a towel with a 50+% average success rate, given less than 10 minutes of human demonstrations (see Figure 1), which is nearly double the success rate compared to CLIP features. Overall, on the basis of these results, we believe that R3M has the potential to become a standard vision model for robot manipulation, which can be simply downloaded and used off-the-shelf for any robot manipulation task or environment. See `https://sites.google.com/view/robot-r3m` for pre-trained models and code.

## 2  Related Work

**Representation Learning for Robotics.** Our work is certainly not the first to study the problem of learning general representations for robotics. One line of work focuses on learning representations from *in-domain* data, that is, using data from the target environment and task for training the representation. Such methods include contrastive learning with data augmentation [25, 26, 27, 28], dynamics prediction [29, 30], bi-simulation [31], temporal or goal distance [32, 33], or domain specific information [34]. However, because they are trained on data exclusively from the target domain and task, the learned representations fail to generalize and cannot be re-used to enable faster learning in unseen tasks and environments.

Recently, there has been growing interest in learning more general representations for motor control from large-scale out-of-domain data like images from the web. This includes the use of CLIP, supervised MS-COCO, supervised ImageNet, MoCo ImageNet features, or data from different robots [35, 36, 37, 38, 23, 39]. In contrast to prior work, we pre-train the representation using diverse human video and language data, as opposed to static frames and/or class labels. Further, in our experimental evaluation, we observe that our pre-trained representation outperforms prior work significantly on a comprehensive evaluation suite. Concurrently, Xiao et al. [40] also explore the use of human interaction data to pre-train visual representations for motor control. However their learned representation only uses static frames from these videos and does not utilize temporal or semantic information like R3M. Furthermore, our evaluation focuses on data efficient imitation learning, and enables real-world learning in cluttered environments with just $\sim 10$ minutes of demonstration data.

**Leveraging Human Videos for Robot Learning**. Several prior works have explored using human video data in robot learning, for example to acquire goals [41, 42, 43], to learn visual dynamics models [44, 45, 46, 47], or to learn representations and rewards [19, 48, 49, 50, 51, 52]. However, these prior works typically focus on a small dataset of human videos closely resembling the robot environment. In contrast, our work leverages diverse human video data like Ego4D [16] to learn visual reusable visual representations that generalize broadly.

**Natural Language and Robotic Manipulation**. Prior works have explored the use of natural language in robot manipulation, primarily as a means of task specification [53, 54, 36, 55] or reward learning [56]. In contrast, we use diverse human video data and language annotations to learn reusable visual representations for control. Prior work has also found visual representations informed by language, like CLIP [12], to be effective for control [36, 37]. Through empirical evaluations, we find that our R3M representation substantially outperforms CLIP for robot manipulation.

**Learning from Diverse Robot Data.** Towards robots that generalize more broadly, there are a number of works that study how to scale up the size and diversity of data robots learn from. Many of these works focus on collecting and learning from robot data itself [57, 58, 7, 8, 9, 10, 59]. However, these works often contain at most a handful of different environments, making generalization across a range of unseen scenes difficult. While we also aim to enable generalization by learning from diverse data, our focus is instead on (1) learning from human video data and hence a larger distribution of environments and tasks, and (2) pre-training a visual representation, as opposed to policies or models.

**Representation Learning from Videos.** Finally, there is a rich literature of works that study learning image representations from videos [60, 61, 19, 62, 63, 64] outside of the context of robotics. Additionally, there are a number of works that use language to learn representations from videos [65, 66]. Critically, unlike all of these works, the main contribution of this work is not to propose a novel representation learning approach, but rather in studying if representations trained on diverse video and language of human interaction can enable more efficient learning of robotic manipulation.

## 3    R3M: Reusable Representations for Robotic Manipulation

Our goal is to use diverse human video data to pre-train a single reusable visual representation for motor control, particularly robotic manipulation, that can enable efficient downstream learning in previously unseen environments and tasks. In this section, we cover the different components of our approach, beginning by describing our problem formulation in Section 3.1, the data sources we use in Section 3.2, and our training objective in Section 3.3.

### 3.1    Preliminaries

Formally, we assume that we have access to a dataset $\mathcal{D}$ of $N$ videos, where each video consists of a sequence of RGB frames $[I_0, I_1, ..., I_T]$. Additionally, we assume that each video is paired with a natural language description $l$, that describes what task is being completed in the video. From this data, our goal is to learn a single image encoder $\mathcal{F}_\phi$, that maps images to a deterministic, continuous embedding, that is $z = \mathcal{F}_\phi(I)$. Once trained, we want to be able to repeatedly reuse $\mathcal{F}$ for downstream policy learning. Specifically, the downstream problem will involve an agent sequentially

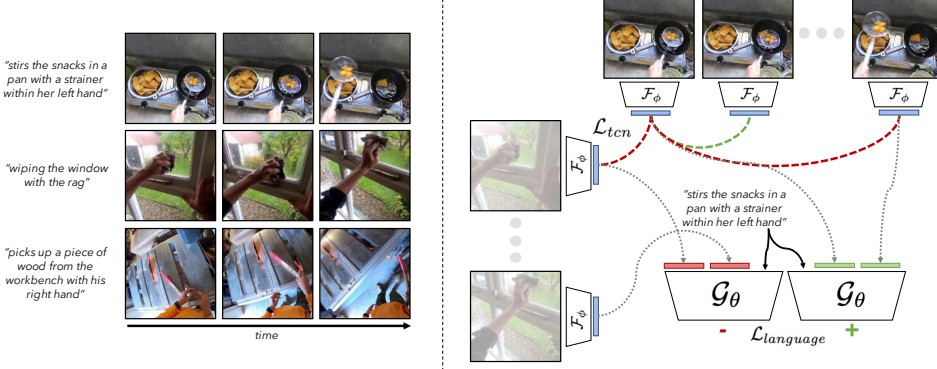

Figure 2: **Ego4D [16] Video and Language (left)**. Sample frames and associated language from Grauman et al. [16] used for training R3M. **R3M Training (right)**. We train R3M with time contrastive learning, encouraging states closer in time to be closer in embedding space and video-language alignment to encourage the embeddings to capture semantically relevant features.

choosing actions given image observations $I$, and instead of using raw images as input, the agent will use the pre-trained $\mathcal{F}_\phi(I)$ as a state representation.

## 3.2 Data Sources

For our learned representation $\mathcal{F}_\phi$ to be useful in a wide range of downstream tasks and environments, it should (1) be trained on data that is diverse enough to facilitate generalization, and (2) provide a useful signal for features relevant to robotic manipulation. One approach would be to be use natural images off the web (e.g. ImageNet [2]). While diverse, these images tend to focus on one particular object, and do not capture an agent interacting with multiple objects in a scene. Alternatively, data of humans interacting in the world [14, 65, 16] is both diverse and contains useful interaction in scenes similar to those we would like robots to interact in. Of the many human video datasets, we leverage the Ego4D dataset [16] due to it's diversity and size, although in principle our method can be used on any suitable video dataset. Ego4D contains videos of people engaging in a wide range of tasks from cooking to socializing to assembling objects from more than 70 locations across the globe, and in total contains more than 3500 hours of data. Each video clip also contains a natural language annotation describing the behavior of the person in the video (See Figure 2 (left)).

## 3.3 Training R3M

What should a good representation for robotic manipulation from human video data capture? We propose three key components: (1) it should capture temporal dynamics, as the agent will be sequentially interacting in the environment to accomplish tasks, (2) it should capture semantically relevant features, and (3) it should be compact. We next describe how we use time contrastive learning to capture (1), video-language alignment for (2), and the use of L1 regularization to encourage (3). See Figure 2 (right) for an overview of our training objective.

**Time Contrastive Learning.** To encourage $\mathcal{F}_\phi$ to capture features relevant to physical interaction and sequential decision making, the first part of our objective is a time contrastive loss [61]. Given a batch of videos we train the encoder to produce a representation such that the distance between images closer in time is smaller than for images farther in time or from different videos. Specifically, we sample a batch of sequences of frames $[I_i, I_{j>i}, I_{k>j}]^{1:B}$, then minimize the InfoNCE loss [67]:

$$\mathcal{L}_{tcn} = -\sum_{b \in B} \log \frac{e^{\mathcal{S}(z_i^b, z_j^b)}}{e^{\mathcal{S}(z_i^b, z_j^b)} + e^{\mathcal{S}(z_i^b, z_k^b)} + e^{\mathcal{S}(z_i^b, z_i^{\neq b})}} \tag{1}$$

where $z = \mathcal{F}_\phi(I)$, and $z_i^{\neq b}$ is a negative example sampled from a *different video* in the batch. $\mathcal{S}$ denotes a measure of similarity, which in our case is implemented as the negative L2 distance.

**Video-Language Alignment.** To encourage $\mathcal{F}_\phi$ to capture semantically relevant features, we train a language prediction module from the embedding outputted by $\mathcal{F}_\phi$. Essentially, by capturing features

predictive of language, like "putting the apple on the plate", the learned representation should capture semantically relevant parts of the scene like the plate and apple state, that are likely relevant to downstream manipulation tasks. Following Nair et al. [56], we train a model $\mathcal{G}_\theta(\mathcal{F}_\phi(I_0), \mathcal{F}_\phi(I_i), l)$ that takes in an initial image $I_0$, a future image $I_i$, language $l$ and outputs a score corresponding to if transitioning from $I_0$ to $I_i$ completes the language $l$. We train the model under the objective that (1) the score should increase over the course of the video, and (2) the score should be higher for correct pairings of video/language than for incorrect pairings. Again we sample a video clip and paired language $[I_i, I_{j>i}, l]^{1:B}$, and then train for this objective directly with a contrastive loss, that is:

$$\mathcal{L}_{language} = -\sum_{b \in B} \log \frac{e^{\mathcal{G}_\theta(z_0^b, z_{j>i}^b, l^b)}}{e^{\mathcal{G}_\theta(z_0^b, z_{j>i}^b, l^b)} + e^{\mathcal{G}_\theta(z_0^b, z_i^b, l^b)} + e^{\mathcal{G}_\theta(z_0^{\neq b}, z_{j>i}^{\neq b}, l^b)}} \tag{2}$$

where again $z = \mathcal{F}_\phi(I)$, and $z^{\neq b}$ is a negative example sampled from a *different video* in the batch (that does not match the language instruction $l^b$).

**Regularization.** Finally, we hypothesize that sparse and compact representations benefit control, particularly in low data imitation learning. State-distribution shift is a well studied failure mode in imitation learning [68], where policies trained with behavior cloning drift off the expert state distribution. Reducing the effective dimensionality of the state space (which we implement with a simple L1 and L2 penalty) can help mitigate this issue, as we demonstrate in Section 4.4.

**R3M Summary & Implementation.** The final objective for training R3M is the weighted sum:

$$\mathcal{L}(\phi, \theta) = \mathbb{E}_{I_{0,i,j,k}^{1:B} \sim \mathcal{D}}[\lambda_1 \mathcal{L}_{tcn} + \lambda_2 \mathcal{L}_{language} + \lambda_3 ||\mathcal{F}_\phi(I_i)||_1 + \lambda_4 ||\mathcal{F}_\phi(I_i)||_2] \tag{3}$$

In principle, R3M can be implemented on top of any encoding architecture for $\mathcal{F}_\phi$. In our experiments we focus on the ResNet50 architecture, and we release pre-trained R3M models with ResNet18, ResNet34, and ResNet50 architectures [69], as well as the accompanying training code. During training, $\phi$ and $\theta$ are trained with an Adam optimizer to minimize Equation 3. Lastly, R3M also trains with random cropping, applied at the video level (that is, within a batch all frames from the same video are cropped identically). Please see the appendix for further implementation details.

# 4 Experiments

In our experiments, we aim to study how the pre-trained R3M representation can be re-used for multiple downstream robot learning tasks. **First**, we study if R3M enables more data efficient imitation learning on unseen environments and tasks compared to existing visual representations and learning from scratch. **Second**, again in the data efficient imitation learning setting, we ablate the different components of the R3M training objective and observe that all components are important for final performance. **Third**, we study if R3M can enable efficient real robot learning in a visually rich household setting. **Finally**, in the appendix, we take a deeper look at task performance of R3M and prior methods with different amounts of data, different camera viewpoints, and different tasks.

## 4.1 Imitation Learning Evaluation Framework

Our evaluation methodology is loosely inspired by Parisi et al. [23]. We focus on evaluating visual representations as frozen perception modules for downstream policy learning with behavior cloning. Given a pretrained visual representation $\mathcal{F}_\phi$, we form the state representation as a concatenation of the visual embedding $z_t = \mathcal{F}_\phi(I_t)$ and the robot proprioceptive (e.g. joint positions and velocities) reading $p_t$. The policy, $\pi$, is trained with a standard behavior cloning loss $||a_t - \pi([z_t, p_t])||_2^2$. We parameterize $\pi$ as a two-layer MLP preceded by a BatchNorm at the input. We train the agent for 20,000 steps, evaluate it online in the environment every 1000 steps, and report the best success rate achieved. For each visual representation and each task, we run 3 seeds of behavior cloning. The final success rate reported on a task is the average over multiple seeds, viewpoints, and demo dataset sizes.

**Comparisons and Baselines.** We compare our **R3M** model to three existing visual representations that have been shown to be effective for control: **CLIP** [12] which trains image representations to be aligned with paired natural language through contrastive learning and has been shown to be

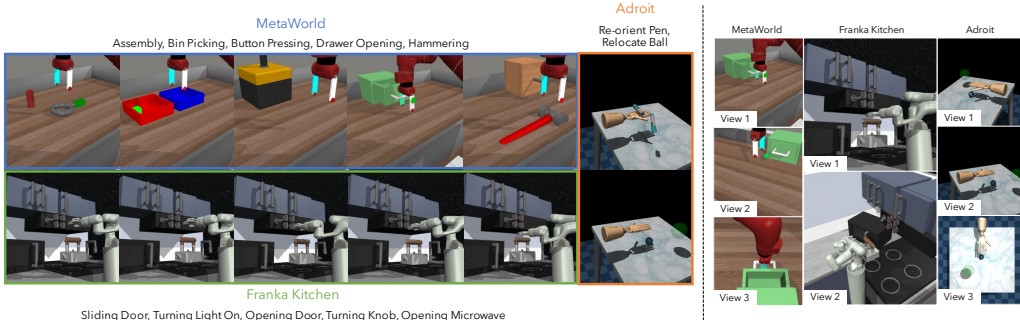

Figure 3: **Simulated Evaluation Environments**. We consider a comprehensive set of manipulation tasks in simulation (**left**), including 5 tasks with a Sawyer from MetaWorld [22], 5 tasks from a Franka operating over a Kitchen [21], and 2 dexterous manipulation tasks from Adroit [20], with multiple views per environment (**right**).

useful for some manipulation [36] and navigation tasks [37], **ImNet Supervised** which uses features pre-trained for ImageNet classification task [2] and has been shown to be effective for reinforcement learning [38], and **MoCo (345) (PVR)** [23], which compresses and fuses the third, fourth, and fifth convolutional layers of a ResNet-50 model trained with MoCo [24] on ImageNet, and has been shown to be effective for imitation learning [23]. We note here that our usage of the Moco (345) model differs from the setup in Parisi et al. [23] in aspects like propreoception features, frame stacking etc. As a result, the numerical results are not directly comparable across the two works. At the same time, we emphasize that all visual representations are used in the same way within our evaluation protocol.

## 4.2   Simulation Environments

Next, we describe the environments and tasks used in our evaluations. For a comprehensive evaluation, we use three robot manipulation domains: MetaWorld [22], the Franka Kitchen environment [21], and Adroit [20] (See Figure 3). Note these environments are only used for downstream learning, and *these environments and tasks are never seen during R3M training*. In the MetaWorld environment we consider the tasks of assembling a ring onto a peg, picking and placing a block between bins, pushing a button, opening a drawer, and hammering a nail. In Franka Kitchen, we learn the tasks of sliding the right door open, opening the left door, turning on the light, turning the stove top knob, and opening the microwave. Finally, in Adroit we consider the tasks of reorienting the pen to the specified position, and picking and moving the ball to specified position. In all tasks, the agent is provided with image observations, as well as proprioceptive data of the robot (end-effector pose, joint positions, etc.) that is concatenated to the encoded image. All tasks involve variation, either by varying the position of the target object in MetaWorld, the positioning of the desk in Franka Kitchen, or the chosen goals in Adroit. For a robust evaluation, we consider multiple views for each environment (See Figure 3), and 3 dataset sizes: $[5, 10, 25]$ in MetaWorld and Franka Kitchen, and $[25, 50, 100]$ in the more challenging Adroit environments. Our comparisons measure performance for each environment and task, averaged over view, dataset size, and object or goal positions.

## 4.3   Exp. 1: Does R3M enable efficient imitation on unseen environments and tasks?

In this first experiment, we measure the success rate of downstream imitation learning using different visual representations. In Figure 4, we first notice that R3M is overall able to learn these vision based manipulation tasks in an extremely low data regime with $\approx 62\%$ success rate, despite never seeing any data from the target environments in training the representation, while outperforming learning from scratch by more than $20\%$. Moreover, we observe that R3M outperforms all prior representations by more than $10\%$ on average across all 12 tasks. By training on diverse interactive video data, and with objectives that capture temporal structure and language relevance, R3M is the best performing method in all 3 environments, and on 11/12 of the tasks (See appendix for performance breakdown by task). The best two performing comparisons are CLIP and MoCo (345) (PVR), with CLIP performing better on MetaWorld, and MoCo (345) (PVR) performing better on Franka Kitchen and Adroit. Unsurprisingly, learning from scratch performs poorly in the low-data regime we study. Ultimately, we conclude that pre-trained visual representations are essential to good performance in the low-data

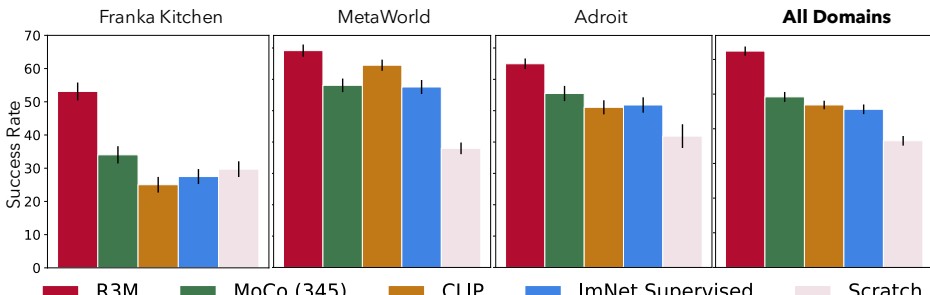

Figure 4: **Data Efficient Imitation Learning in Unseen Environments/Tasks.** We report the success rates of downstream imitation learning with standard error bars. We observe that across 12 tasks R3M outperforms baselines like MoCo (345) (PVR), CLIP, Supervised ImageNet features, and training from scratch.

imitation learning regime, and using R3M with diverse human video data is especially effective for learning representations useful for robotic manipulation.

## 4.4  Exp. 2: Which components of R3M are important?

In this experiment, we seek to understand the different components of R3M, beginning with the objective. Specifically, we compare the full **R3M** with **R3M(-Aug)**, which does not use crop augmentations, **R3M(-L1)**, which does not include $L1$ regularization, and **R3M(-Lang)**, which does not include include the video-language alignment loss.

| Environment | R3M | Supervised R3M(-Aug) | R3M(-L1) | Self-Supervised R3M(-Lang) |
|---|---|---|---|---|
| Franka Kitchen | **53.1** ±2.7% | 51.1 ±2.7% | 46.7 ±2.7% | 47.2±2.9% |
| MetaWorld | **69.2** ±2.0% | 68.9 ±2.1% | 65.0 ±2.4% | 67.0±2.0% |
| Adroit | 65.0 ±1.7% | 61.3 ±2.1% | **66.5** ±1.6% | 45.6 ±3.3% |
| All Domains | **62.4** ±1.3% | 60.4 ±1.4% | 59.4 ±1.5% | 53.2 ±1.5% |

Table 1: **Ablating Components of R3M**. We see report success rate of downstream imitation learning on variants of R3M. We observe that on average, removing the L1 penalty have a negative impact, particularly on the Franka Kitchen and MetaWorld environments. Lastly, removing language grounding has the most significant drop in performance, particularly on the Adroit tasks.

In Table 1, we report success rates per environment and averaged over all environments. First, we notice that on average across the three environments, we see a drop in performance of ≈2% from removing crop augmentation or from removing the $L1$ regularization. Interestingly, the impact of removing the sparsity regularization depends on the environment. In Franka Kitchen and MetaWorld, sparsity is helpful, while in Adroit removing sparsity actually helps performance slightly. We suspect this is partly due to the Adroit environment using more demonstrations, mitigating the state distribution shift issue.

We see that across all environments, removing video-language alignment loss has the largest negative impact on performance, particularly in the Adroit environment. We hypothesize that language alignment plays an important role in better capturing semantic features that might be predictive of objects and useful for object manipulation. Nevertheless, we note that even in the fully self-supervised regime, our R3M model still outperforms prior state of the art visual representations like ImageNet trained MoCo (345) (PVR) [23] and CLIP [12] by a significant margin.

Next, we seek to answer the question: *How important is the data?* To do so we include comparisons that disentangles the role of the dataset and the training objective. In particular, we have trained a MoCo model on the exact same frames of the Ego4D dataset used to train our R3M model (See Table 2). Additionally we compare to the MVP model [70], which trains a ViT-B masked auto-encoder on the Ego-soup dataset, which comprises of Ego4D and

|  | Franka | Adroit |
|---|---|---|
| R3M | **53.1**(2.7) | **65.0** (1.7) |
| MoCo-Ego4D | 42.0 (2.8) | 54.9 (2.7) |
| MVP ([70]) | 27.0 (2.6) | 51.4 (2.7) |

Table 2: **Importance of Data vs. Algorithm**. We find that the MoCo-Ego4D and MVP models, which leverage the same or more data and compute as R3M perform more than 10% worse.

other egocentric video datasets.. We evaluate these comparisons on the Franka Kitchen and Adroit environments, and find that the **MoCo-Ego4D model, which uses the same data and compute as R3M, gets an average success rate ~10% lower than R3M in both environments**. Moreover, we find the MVP models performs ~20% worse than R3M. This suggests that while there is indeed a

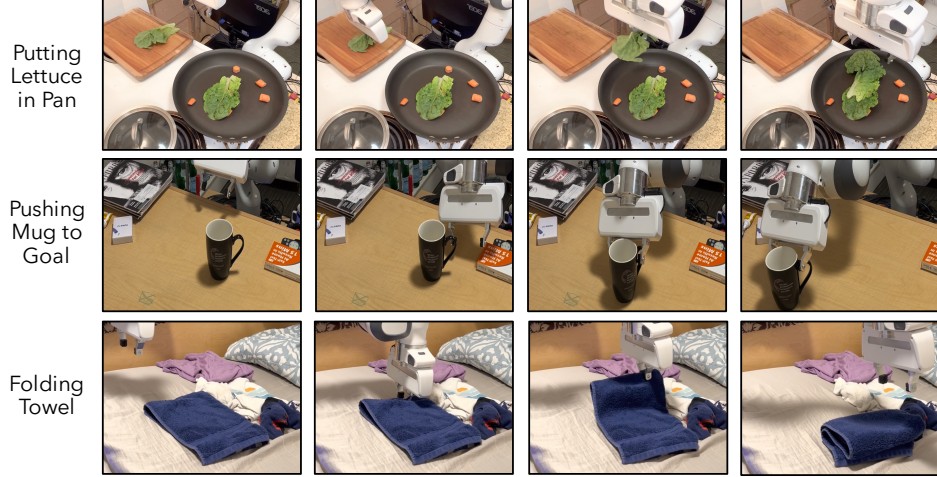

Figure 5: **Real World Robot Learning with R3M.** With R3M we are able to learn challenging tasks like putting lettuce in the pan, pushing the cup to the goal, and folding the towel from just 20 demonstrations. See appendix for more examples of real robot tasks and details about the robot setup.

large benefit coming from diverse human video data compared to static ImageNet images (34% → 42% on Franka), the data is not the only source of improvement, and the R3M objective provides an additional ∼ 10% boost in success rate.

### 4.5 Exp. 3: Does R3M enable data efficient learning in real world environments?

Finally, we test if R3M can enable data-efficient robot learning in cluttered real-world environments. To do so, we bring a Franka Emika Panda robot into a real graduate student apartment, and aim to learn household tasks from pixels with just 20 demonstrations per task, using the pre-trained R3M representation. We have the robot complete five tasks: (1) closing a dresser drawer, (2) picking a face mask placed randomly on a desk and placing it in the dresser drawer, (3) picking up lettuce randomly placed on a cutting board and putting in a cooking pan, (4) pushing a mug to a goal location, and (5) folding a towel (See Figure 5). Like in our simulation experiments, we collect a small number of demonstrations and do simple behavior cloning with the pre-trained representation.

In Table 3, we report the success rates comparing R3M and CLIP, one of the stronger baselines from our evaluations in simulation. We observe that while the two perform similarly on the easier task of closing the drawer, R3M consistently performs better on the other four tasks (See Figure 5), which require more precise visual representations, yielding nearly double the success rate on average.

| Success out of 10 trials | R3M | CLIP |
| --- | --- | --- |
| Closing Drawer | **80%** | 70% |
| Putting Mask in Dresser | **30%** | 10% |
| Putting Lettuce in Pan | **60%** | 0% |
| Pushing Mug to Goal | **70%** | 40% |
| Folding Towel | **40%** | 0% |
| Average | **56%** | 24% |

Table 3: **Real World Success Rates.** R3M outperforms CLIP on the challenging real world manipulation tasks.

## 5 Limitations and Future Work

In this work, we set out to study if pre-training visual representations on diverse human videos can enable efficient learning of downstream robotic manipulation tasks. While we were excited by strong results on a wide set of simulated and real robotic tasks, a number of important limitations remain. Our current evaluation is limited to imitation learning, specifically behavior cloning, with a small number of task demonstrations. While we would hope to see R3M be equally beneficial for other robotic learning settings like reinforcement learning, it could be the case that a good pretrained representation for RL is not the same as a good pre-trained representation for imitation. Studying how R3M performs in RL settings, and changes that may need to made to improve its performance is an exciting next step. The current R3M model also only provides a single-frame state representation. In principle, pre-training on human videos should be able to go beyond state representations (e.g. reward learning and task specification). Studying if R3M embeddings or the language grounding module can provide a useful reward signal is an interesting direction for future work.

**Acknowledgments**

The authors would like to thank the Ego4D team at Meta AI for assistance in using the dataset. We'd also like to thank Karl Pertsch, Simone Parisi, Sidd Karamcheti, and numerous members of Meta AI and the IRIS labs for valuable discussions. This work is in part supported by ONR grant N00014-22-1-2621. Finally, the authors would also like to thank Evan Coleman for assistance with the robot.

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

# A R3M Training Details

## A.1 Data Preprocessing

The Ego4D dataset consists of several hour long videos within a certain scene. Within each scene, there are many sub-clips, each with a natural language annotation. R3M trains with these shorter video clips paired with language annotations.

For faster training R3M parses each video clip into frames (Resized and cropped to 224x224) and samples frames from a video clip individually. See the codebase for more details on the implementation of sampling the videos.

## A.2 Training Architecture and Hyper-Parameters

R3M can in principle be trained with any visual encoding architecture for $\mathcal{F}_\phi$. We train with off the shelf ResNet18, 34, and 50 [69], as implemented by `torchvision.models`.

The language prediction head is implemented as an 5 layer MLP with sizes [2*$E$ + $L$, 1024, 1024, 1024, 1024] and output a scalar score, where $E$ is the output dimension of $\mathcal{F}_\phi$ and $L$ is the output dimension of the DistilBERT [71] sentence encoder (768) from HuggingFace `transformers`.

During training of R3M, we use batch sizes of 16 video clips (where 5 frames are samples from each video clip: an initial image, final image, and sequence of 3 frames). The initial and final frames are sampled from the first and last 20% of the video clip.

R3M models are trained for one million steps in our experiments, and for 1.5 million steps in our released models, with a learning rate of 0.0001.

For the training objective in Equation 3, we use hyperparameters $\lambda_1 = 1, \lambda_2 = 1, \lambda_3 = 0.00001, \lambda_4 = 0.00001$.

## A.3 Additional Implementation Details

In practice, we use more than one negative video example in training Equations 1 and 2. Instead we use 3 negative examples, sampled from different videos in the batch.

Additionally in training for Equation 2, we consider the following positive pairs within a single batch element: Initial and Final Frames $(I_0, I_g)$, $(I_0, I_{j>i})$, and $(I_0, I_{k>j})$, with corresponding negatives $(I_0, I_0)$, $(I_0, I_i)$, and $(I_0, I_j)$ respectively. Using a larger number of positive examples from a single video and multiple negative examples from different videos stabilizes training.

## A.4 Example Usage

Using R3M is simple. The codebase is located at https://github.com/facebookresearch/r3m. Simply clone the repo and install via `pip install -e .` Then R3M can be loaded by running:

```
from r3m import load_r3m
r3m = load_r3m("resnet50") # resnet18, resnet34
r3m.eval()
```

# B Evaluation Details

## B.1 Simulation Environments

We focus on three simulation environments: Franka Kitchen, MetaWorld, and Adroit.

**Franka Kitchen**. The Franka Kitchen environments used in this paper are modified from the original environment; specifically, we add additional randomization to the scene. We randomly change the position of the kitchen between episodes, making the task significantly more challenging both in perception and control.

The 5 tasks in the Franka Kitchen involve opening the left door, opening the sliding door, turning on the light, turning the knob, and opening the microwave. All Franka tasks include proprioceptive data

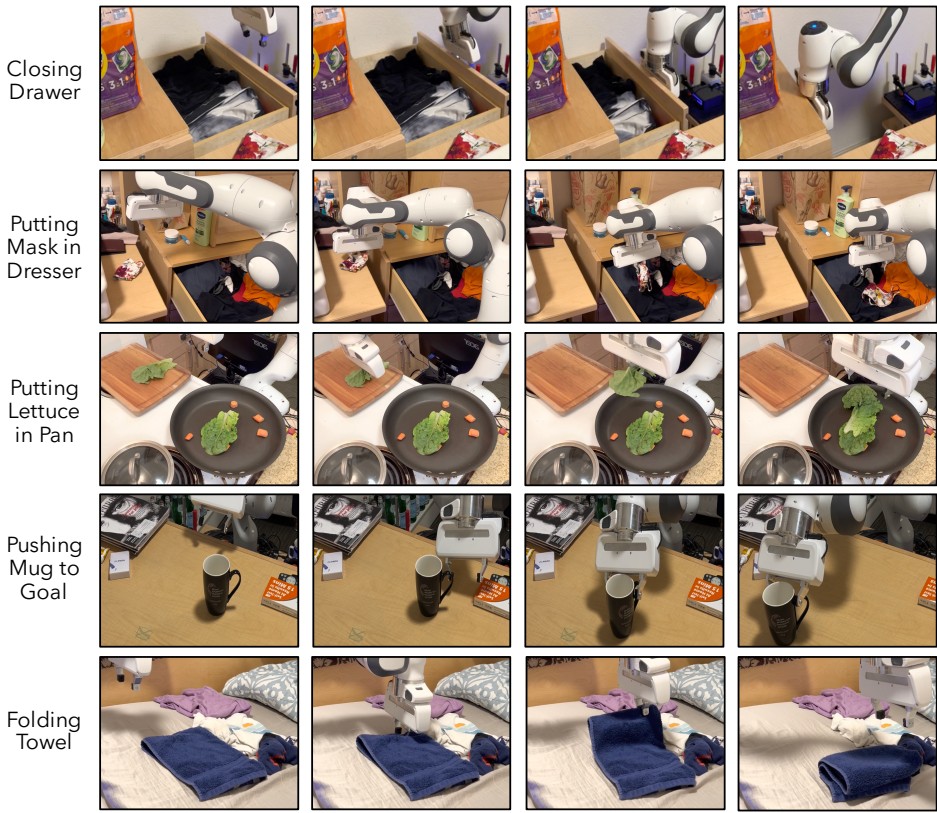

Figure 6: **Real World Robot Learning with R3M.** With R3M we are able to learn challenging tasks like closing the drawer, putting the mask in the dresser, putting lettuce in the pan, pushing the cup to the goal, and folding the towel from just 20 demonstrations.

of the arm joint positions and gripper positions. The horizon for all Franka tasks is 50 steps, and our imitation experiments use either 5, 10, or 25 demos.

**MetaWorld**. The MetaWorld environments are the standard V2 Button Pressing, Bin Picking, Drawer Opening, Hammer, and Assembly environments available in MetaWorld [22]. In all tasks, the target object (drawer, peg, block, etc.) position is randomized between episodes.

All MetaWorld tasks include proprioceptive data of the gripper end effector pose and gripper open/-close. The horizon for all MetaWorld tasks is 500 steps, and our imitation experiments use either 5, 10, or 25 demos.

**Adroit**. We use the standard Pen and Relocate tasks in the Adroit hand manipulation suite. The goal position of the pen and the goal position of the ball are randomized between episodes, and specified visually.

All Adroit tasks include proprioceptive data of the hand joints, and in the Relocate task also includes the global position of the hand. The horizon for the Pen task is 100 steps and for the Relocate task is 200 steps. Our imitation experiments use either 25, 50, or 100 demos.

## B.2 Real World Environments

Our real world experiments involve bringing a Franka Emika Panda robot into a real graduate student apartment. The tasks involve putting lettuce in a pan in the kitchen, pushing a mug to a goal position on a dining table, closing a drawer, putting a mask in a drawer, and folding a towel (See Figure 6). All tasks involve randomization (e.g. the towel/lettuce/mug/mask position or drawer position). The initial state of the gripper is also randomized each episode.

| Folding Towel | Closing Drawer | Putting Mask in Dresser | Pushing Mug to Goal | Putting Lettuce in Pan |

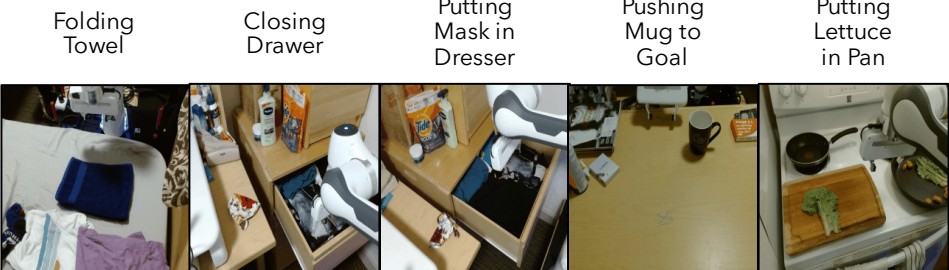

Figure 7: **Real Robot Camera Viewpoints.** Camera view used for learning each of the real robot tasks.

The robot observation includes RGB images froma USB webcam, positioned differently for each task (See Figure 7). The robot end effector position is also concatenated with the image embedding during imitation learning.

### B.3   Demo Data Collection

In the Franka Kitchen and Adroit tasks, expert data is generated by training a state based agent with model free RL [20]. The state based trajectories are then replayed and rendered with image observations.

In the MetaWorld environment, a heuristic policy using state information is used to generate expert data, which is then replayed and rendered with image observations.

On the real robot, demonstrations are collected by a human tele-operator with a PlayStation controller. The control is applied directly in the end effector Cartesian space, and the demo trajectories are directly saved with visual observations.

### B.4   Comparisons

In all experiments all models use a ResNet50 base architecture.

**CLIP**: The CLIP comparison uses the of the shelf CLIP RN50 model available at `https://github.com/openai/CLIP`.

**ImNet Supervised**: This comparison uses the default ResNet architecture available from `torchvision.models` with `pretrained=True`.

**MoCo (345)**: This comparison uses a pre-trained MoCo model on Imagenet which fuses the third, fourth, and fifth convolutional layers as proposed in [23].

Note that our usage of the Moco (345) model differs from the setup in Parisi et al. [23] in aspects like proprioception features, frame stacking etc. As a result, the numerical results are not directly comparable across the two works.

**Scratch**: uses the default ResNet architecture available from `torchvision.models` with `pretrained=False`. Additionally, it lets gradients from the behavior cloning MSE loss pass into the visual encoder.

**MoCo-Ego4D**: This comparison uses a pre-trained MoCo model on the samed data as R3M from the Ego4D dataset.

**MVP**: This comparison uses a pretrained MVP [40, 70] model, which trains an MAE with a ViT-B architecture on the Ego-Soup dataset, which consists of Ego4D and other egocentric human video datasets.

### B.5   Behavior Cloning Hyperparameters

The downstream policy is a 2 layer MLP with hidden sizes [256,256] preceded by a BatchNorm. The input to the policy is the concatenated visual embedding and proprioceptive data, and the output is

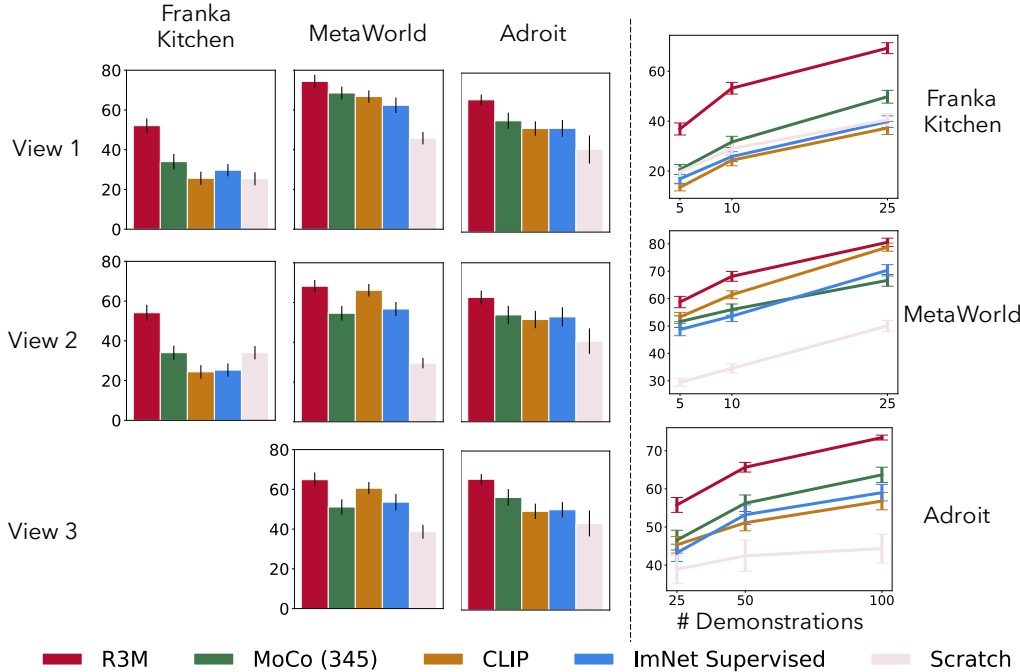

Figure 8: **Performance over different views/dataset sizes**. We report the success rate of R3M and baseline across each view (**left**) and dataset size (**right**). We see that the performance improvement from R3M is consistent across all views. We also observe that while absolute performance increases with more demos, the performance improvement from R3M is consistent across all demo sizes.

the action. The policy is trained with a learning rate of 0.001, and a batch size of 32 for 20000 steps, evaluating every 1000.

## C  Additional Results

### C.1  How does performance vary across viewpoint and demo dataset size?

In our next experiment, we take a closer look at R3M performance compared to prior methods across viewpoints and dataset sizes. In Figure 8, we plot the average success rate of each method across each dataset size and viewpoint. We observe that the performance improvement of R3M is consistent across all viewpoints, and it is the highest performing representation in all cases. Interestingly, we see that the same does not hold amongst the prior methods, where the ranking between MoCo (345) and CLIP changes based on the chosen viewpoint.

Additionally, we also study the impact of dataset size for imitation learning. Again, we observe that the performance improvement from R3M is consistent, outperforming the baselines across every environment and demo dataset size. We observe that in the Franka Kitchen and Adroit environments, the performance gain from R3M stays consistent with increase in dataset size, even as the absolute performance of all methods improves. Overall, we clearly observe that the performance benefit of R3M is not tied to a specific viewpoint or dataset size.

### C.2  Performance Breakdown By Task

In Figure 9 we report the success rate on each task individually. Note each success rate for each method is still the average over 3 views, 3 demo sizes, and 3 seeds. We observe that on 11/12 tasks R3M is the highest performing method.

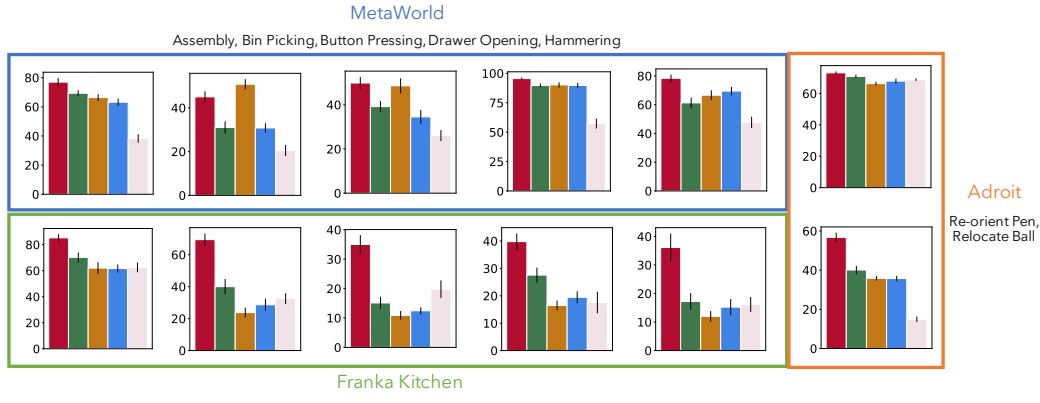

Figure 9: **Per task Success Rate.** We observe that R3M is the highest performing method on 11/12 tasks.

