# OpenReview forum: "R3M: A Universal Visual Representation for Robot Manipulation"
_robot-learning.org/CoRL/2022/Conference — CoRL 2022 Poster_

### Official Review · Reviewer_QF3c · 2022-07-06

**Originality:** Very Good
**Technical Quality:** Good
**Clarity Of Presentation:** Very Good
**Impact:** 4

**Recommendation:**

Strong Accept: I recommend accepting the paper and will argue for my recommendation even if other reviewers hold a different opinion.

**Summary:**

The paper proposes a novel visual representation learning approach for robotic manipulation.
The algorithm uses language annotated human video sets (Ego4D) and contrastive learning to learn a consistent visual representation.
The resulting pre-trained visual representation can then be deployed to downstream robot manipulation tasks.
The promise of R3M is to improve data-efficiency of policy learning by providing a pretrained visual module as opposed to learning it from scratch.
Extensive simulated and real world results show that R3M is able to outperform other methods in learning robot policies from demonstrations via behavior cloning in previously unseen environments.

**Issues:**

Could you put as much results as possible into the main paper from the supplementary material? The figures showing the success rates take up quite some space, I'd recommend using a table instead and also presenting Fig. 8 and 9 from the Supplementary material.

Question: why does the paper emphasize R3M to be 'frozen'. Couldn't we just update it in downstream tasks to fit the task a bit better?



**Quality Of The Limitations Section:**

Limitations are addressed clearly

**Reviewer Expertise:**

4: The reviewer is confident but not absolutely certain that the evaluation is correct

**Robotics Focus:**

Sufficient demonstration on hardware

**Strengths And Weaknesses:**

Strengths

The paper proposes the original idea of using video recordings of human manipulation and interaction tasks together with language annotation to learn a visual representation network.
Due to the diversity of robotic manipulation pre-training datasets are not, or only sparsely available.
However, relying on real-world video recording of human manipulation can form a basis for vision-guided robot manipulation tasks as well.
The paper proposes an intuitive self-supervised learning method that is able to exploit the sequential information in videos, language annotations to capture semantic meaning and enforces sparsity of the image embeddings to avoid overfitting.
The overall method is well motivated and the paper is easy to follow.
Despite the limited page number the paper provides simulated and real robot results in the context of imitation learning with BC using the pre-trained visual representation.

Weaknesses

While the idea is original and promising, a significantly extended evaluation will be needed to fully support the claims of the paper.
The paper also discusses limitations on this aspect.
It would be also interesting to see a deeper analysis on what knowledge from the Ego4D dataset is relevant for the (simulated) robotic tasks.
Overall, a more in-depth qualitative analysis of the approach would be welcome.

**Summary Of Recommendation:**

The paper proposes an original and potentially impactful idea for visual representation in robot manipulation.
While this work could use a significantly extended page number to fully support its claims, the paper at its present state does a good job in conveying the main messages.

---

> ### Author Response · Authors · 2022-08-23
> **Response to Reviewer QF3c**
>
> We thank the reviewer for their insightful comments and positive feedback on the paper.
>
> Regarding the weakness mentioned by the reviewer,
> > It would be also interesting to see a deeper analysis on what knowledge from the Ego4D dataset is relevant for the (simulated) robotic tasks
>
> To provide further insights, we added a new comparison that disentangles the role of the dataset and the training objective. In particular, we have trained a MoCo model on the exact same frames of the Ego4D dataset used to train our R3M model (See Table below). We then evaluated this MoCo-Ego4D model on the Franka Kitchen environment. We find that the MoCo-Ego4D model gets an average success rate of 36.0%, while R3M gets 41.2%. This suggests that while there is indeed a large benefit coming from diverse human video data compared to static ImageNet images (28% → 36%), the data is not the only source of performance improvement, and the R3M objective is particularly effective at learning a representation from such human videos, providing an additional ~5% boost in success rate.
>
> |------------|----------------|----------------|----------------|----------------|----------------|----------------|----------------|----------------|----------------|
>
> |            | Average        | Knob Turn      | Light On       | Sliding Door   | Left Door      | Microwave Open | 5 Demos        | 10 Demos       | 25 Demos       |
>
> | R3M        | **41.2 (2.6)** | **30.1 (1.6)** | **51.3 (2.7)** | **73.3 (2.1)** | **25.1 (1.6)** | **26.1 (2.0)** | **27.0 (2.1)** | **40.5 (2.4)** | **56.0 (2.6)** |
>
> | MoCo-Ego4D | 36.0 (2.4) |  28.3 (1.3) |  47.3 (2.1) | ** 71.6** (1.9)   | 10.7 (0.7) | 21.7 (1.4)   |  24.2 (1.9) |  36.2 (2.4) | 47.8 (2.6)  |
>
> We have added this experiment to the revision.
>
> > Could you put as much results as possible into the main paper from the supplementary material? The figures showing the success rates take up quite some space, I'd recommend using a table instead and also presenting Fig. 8 and 9 from the Supplementary material.
>
> Thank you, this is a great suggestion, we will convert our results to a table and move the results from the appendix into the main paper for the camera ready.
>
> > Question: why does the paper emphasize R3M to be 'frozen'. Couldn't we just update it in downstream tasks to fit the task a bit better?
>
> Absolutely, R3M can be used ‘frozen’ or can easily be finetuned. In our experiments we aim to limit confounding factors when comparing pre-trained representations, hence we do not finetune. However finetuning likely will improve performance even further.

---

### Official Review · Reviewer_QzCR · 2022-07-14

**Originality:** Good
**Technical Quality:** Good
**Clarity Of Presentation:** Good
**Impact:** 3

**Recommendation:**

Weak Accept: I recommend accepting the paper, but will not argue for my recommendation if the majority of other reviewers have a different opinion.

**Summary:**

This paper proposes reusable representation for robotic manipulation (R3M), a framework that leverages time-contrastive learning, video-language alignment, and an L1 penalty to encourage sparse and compact representations, learning reusable representations at a pre-training stage using Ego4D human video dataset. Experiments are ran on the MetaWorld, the Franka Kitchen, and Adroit environments. This is an immitation learning method, requiring annotatated videos with sentences describing tasks.


**Issues:**

I have the following questions and concerns:

- Word-to-word repeat of the following paragraph : `We hypothesize that training a good representation for robotic manipulation from human video data should capture three key components ...` In section 1 Intro and 3.3.
- When using contrastive learning, for negative samples, you sample a negative example sampled from a different video in the batch.
Does different video mean a completely different task? how do you ensure the tasks are non-related so the contrastive learning objective can work best?

- The tasks attempted are slightly non-standard, especially going from the tasks tackled in simulation vs real world. I am wondering how the authors picked these tasks?

- Where is the balance between needing variety of annotations vs tasks to learn meaningful representations? Since Ego4D dataset already contains these, but when you want to learn the tasks on the real system? For instance do you need one sentence per task, or variety of tasks? This was briefly discussed in the CARE paper (referenced above) where there is heavy reliance on the exitence of language annotations.

**Quality Of The Limitations Section:**

Limitations are addressed clearly

**Reviewer Expertise:**

5: The reviewer is absolutely certain that the evaluation is correct and very familiar with the relevant literature

**Robotics Focus:**

Sufficient demonstration on hardware

**Strengths And Weaknesses:**

## Strengths
- The usage of large-scale datasets to build reusable representations in RL and robotics in particular, as opposed to training tabula-rasa, is a well-motivated research problem.
- I agree with the authors to certain  degree that one cause for the representation component of RL behind behind of other ML problems such as NLP can be due to our lack of pre-training strategy agreements and access to appropriate robotics datasets.
- Compelling results and real world robot results in general.


## Weaknesses
- The reliance on annotated dataset, where each video is paired with a natural language description, that describes what task is being completed in the video.
- No other baseline presented, other than using their framework emebedded with representation learning techniques such as CLIP and MoCo. This makes the claims weak, since there are other frameworks focused on representation learning for immitation learning/RL. Other baselines to consider could be direct comparison to PVR, or CURL (contrastive-based learning) / other immitation learning frameworks etc.
- Weak contribution/novelty since the method is combining a few existing pieces under an immitation learning framework where in general its less of a challenging setting compared to RL. Also experimenting with regularization methods such as $L1$ is a common practice, for me it doesn't add any value in terms of contribution (its more like an implementation detail).
- Language embedding for representations is not new, for instance CARE (https://arxiv.org/pdf/2102.06177.pdf) is one example.

**Summary Of Recommendation:**

The main components of this method are as follows:

Starting with dataset $D$ of $N$ videos, where each video consists of a sequence of RGB frames. Ego4D dataset (tasks such as cooking to socializing to assembling objects from more than 70 locations across the globe).


### Time Contrastive Learning.
Given a batch of videos, they train the encoder to produce a representation such that the distance between images closer in time is smaller than for images farther in time or from different videos.

### Video-Language Alignment.
In order to capture semantically relevant features, they train a language prediction module from the embedding. That is, capturing features
predictive of language, like “putting the apple on the plate”.

### Regularization
L1 regularization to encourage learning sparse representations.

### Experiments:
In the MetaWorld environment the tasks are assembling a ring onto a peg, picking and placing a block between bins, pushing a button, opening a drawer, and hammering a nail.
In Franka Kitchen, sliding the right door open, opening the left door, turning on the light, turning the stove top knob, and opening the microwave. In Adroit, reorienting the pen to the specified position, and picking and moving the ball to specified position.


Overall this paper demonstrates interesting arguments and a variety of interesting manipulation tasks - But the core contributions are not compelling. I am happy to discuss and change my score if provided with a strong argument for the lack of external baseline methods. More in particular, I'd like to see comparisons with other SOT of immitation learning methods.

---

> ### Author Response · Authors · 2022-08-23
> **Response to Reviewer QzCR (1 of 2)**
>
> We thank the reviewer for their detailed and helpful feedback. Below we address each comment, and make a number of important clarifications that we believe may address the reviewers concerns.
>
> > The reliance on annotated dataset, where each video is paired with a natural language description, that describes what task is being completed in the video.
>
> In Table 1 we show the performance of R3M (-Language), which is a fully self-supervised version of R3M without the reliance on an annotated dataset. This ablation still outperforms all baselines by a statistically significant margin. So while we get the best performance by using language, even without an annotated dataset R3M still outperforms state of the art models.
>
> > No other baseline presented, other than using their framework emebedded with representation learning techniques such as CLIP and MoCo. This makes the claims weak, since there are other frameworks focused on representation learning for immitation learning/RL. Other baselines to consider could be direct comparison to PVR, or CURL (contrastive-based learning) / other immitation learning frameworks etc.
>
> **This is an important misunderstanding**. We clarify that **we do include a head-to-head comparison to PVR (Parisi et al)**. The model labeled “MoCo (345)” is the exact model from the PVR paper, as their contribution was in fusing the 3rd, 4th, and 5th layers of a network trained with MoCo. We hope that this addresses the reviewers concern regarding comparison to the state of the art.
>
> > Weak contribution/novelty since the method is combining a few existing pieces under an immitation learning framework where in general its less of a challenging setting compared to RL. Also experimenting with regularization methods such as L1 is a common practice, for me it doesn't add any value in terms of contribution (its more like an implementation detail).
>
> R3M is the first paper to leverage diverse human video datasets for pre-training visual state representations for robotic manipulation. In doing so, our core contribution is an artifact – the pre-trained vision model – that can be used readily in other work. Our main contribution is not a new representation learning algorithm. We have revised the introduction of the paper to make this clearer. We believe that leveraging human videos is an essential insight and advancement over prior work in pre-training visual representations for control, which utilize static image data (RRL [1], PVR [2], CLIPort [3], EmbCLIP [4]). R3M outperforms all of these methods in a head-to-head comparison on a comprehensive simulated and real robot evaluation. In fact, there are at least 4 CoRL '22 submissions that use the R3M pre-trained model, all of which are not affiliated with us. We exclude links to these submissions to preserve anonymity.
>
> [1] Shah and Kumar. RRL: Resnet as representation for Reinforcement Learning. ICML 2021.
> [2] Parisi et al. The (Un)Surprising Effectiveness of Pre-Trained Vision Models for Control. IMCL 2022.
> [3] Shridar et al. CLIPort: What and Where Pathways for Robotic Manipulation. CoRL 2021.
> [4] Khandelwal et al. Simple but Effective: CLIP Embeddings for Embodied AI. CVPR 2022.
>
> > Language embedding for representations is not new, for instance CARE (https://arxiv.org/pdf/2102.06177.pdf) is one example.
>
> This paper is absolutely relevant; we will cite and discuss it in the revision. We emphasize that the contribution of R3M is not in a new representation learning algorithm for vision and language, but rather in leveraging large-scale human videos to produce a reusable representation for robotic manipulation. Additionally, unlike CARE, the R3M representation is purely a visual one, and is not conditioned on language during inference and policy training.
>
> > Word-to-word repeat of the following paragraph : We hypothesize that training a good representation for robotic manipulation from human video data should capture three key components ... In section 1 Intro and 3.3.
>
> Thank you for pointing this out; we have corrected this in the revision.

---

> ### Author Response · Authors · 2022-08-23
> **Response to Reviewer QzCR (2 of 2)**
>
>
> > When using contrastive learning, for negative samples, you sample a negative example sampled from a different video in the batch. Does different video mean a completely different task? how do you ensure the tasks are non-related so the contrastive learning objective can work best?
>
> Given a sufficiently large and diverse dataset like Ego4D, when randomly sampling a negative video from the batch, the probability of sampling the same task is incredibly low. So while a random sample has some very small chance of being the same/related task, we find in practice this does not impact learning.
>
> > The tasks attempted are slightly non-standard, especially going from the tasks tackled in simulation vs real world. I am wondering how the authors picked these tasks?
>
> In simulation we aimed to have a diverse evaluation that measured representation quality across a wide range of manipulation tasks, from opening cabinets/microwaves to rearranging objects to in-hand manipulation. Hence we considered 3 popular and standard benchmarks (Franka Kitchen, MetaWorld, and Adroit), and multiple representative tasks from each. The benchmark tasks were all pre-defined before running evaluations and not cherry picked in favor of R3M.
>
> In the real world, we aimed to tackle realistic household tasks with visual complexity, and therefore took our Franka robot into a graduate student apartment and again studied household tasks like rearranging objects, folding towels, and opening/closing drawers. We specifically chose tasks not too difficult for imitation learning (e.g. “making coffee from scratch”), but with sufficient visual complexity to test the real world applicability of the pre-trained visual model.
>
> > Where is the balance between needing variety of annotations vs tasks to learn meaningful representations? Since Ego4D dataset already contains these, but when you want to learn the tasks on the real system? For instance do you need one sentence per task, or variety of tasks?
>
> We clarify that language is only used once during pre-training on the large-scale human video dataset; after the representation is pretrained, language/annotations are no longer needed for the downstream tasks. The one pre-trained encoder can be reused for many downstream tasks, and in our experiments just one single pre-trained R3M model is used for all tasks in the paper.

---

### Official Review · Reviewer_JMfs · 2022-07-30

**Originality:** Fair
**Technical Quality:** Good
**Clarity Of Presentation:** Good
**Impact:** 3

**Recommendation:**

Strong Reject: I recommend rejecting the paper and will argue for my recommendation even if other reviewers hold a different opinion.

**Summary:**

The paper proposes to use a deep neural network model trained on Ego4D human videos to generate latent representations (visual embeddings) with the hypothesis that it will speed up learning as compared to training from scratch or using other visual embedding generation methods.

**Issues:**

The manuscript requires a significant re-write with an emphasis on analysis and a deeper dive on the fundamentals of why R3M works (slightly) better than existing methods with concrete statements and experiments to verify. It may be ambitious to attempt this during the short few weeks of rebuttal. The issues are listed in sections above.

**Quality Of The Limitations Section:**

Limitations are not well addressed

**Reviewer Expertise:**

5: The reviewer is absolutely certain that the evaluation is correct and very familiar with the relevant literature

**Robotics Focus:**

Relevant but unlikely to deploy to hardware in near future

**Strengths And Weaknesses:**

Merits and Strengths:

The paper does indeed address an important challenge in manipulation: what should be the backbone model/representation for the robot, given that it is so expensive to collect data? The contrast between robotics and CV/NLP is also drawn where the latter fields have benefited greatly from LLM and VLMs trained on large corpi of data – can we benefit from that in robotics?

The paper does a good job of evaluating and benchmarking the proposed approach on a number of simulated environments. There are also a number of robot experiments implemented with a Franka Emika Panda in the context of in-home robotic applications. The benchmarks mostly seem appropriate.

Weaknesses and Limitations:

The biggest concern with the paper is: What is the core technical contribution/novelty? The paper uses an existing data-set, together with a well-established learning procedure (the losses, data preparation, and model are fairly standard) to generate an encoder that is then used in a fairly standard imitation learning framework. It seems that the essence of the paper is “using a model trained on Ego4D data set may be better than using one trained on CLIP or other similar data-sets”. I expected to see some contribution along some axes such as: insight for general principles of what makes a data-set particularly useful for robotic application, or some novelty in training, data-augmentation, or generalization. For example, what specifically about the R3M data-set makes it more effective than CLIP , ImNet, or MoCo? What types of data should we be collecting based on the preliminary results of this manuscript? The experiments should be more specific to tease out these differences and highlight why R3M is able to outperform, not just that it does.

Beyond the above concern, the performance improvement is quite modest. The success rate in the Franka Kitchen environment is low for all models, and there is a modest (about 15% based on the graph) improvement but nothing to suggest that R3M is the solution. The Meta Worlds results are inconclusive, the performances are quite similar. Adroit is between the two, R3M does better but it's within 10% of the next baseline. We cannot conclude that R3M is a significant advance on the state-of-the-art based on these results.

The limitations discussed are okay. I agree with the first one, it is not obvious whether a representation used for imitation learning is also good for RL. The second limitation discussed is more of a future work rather than an issue with the approach. The model provides a functionality (language module) that was not used in this study but could potentially be useful for other tasks. I find the last limitation the least satisfying, it is basically a statement of “more data is probably better”. That statement is likely true, but this is a limitation of all learning methods, the more data the better they are likely to perform.


**Summary Of Recommendation:**

The manuscript provides some encouraging preliminary results; however, it needs to provide deeper insights into why the particular model works (what is specifically useful about the data-set) and what general principles can we garner/look into studying to improve our visual encoders for robotics applications (e.g., what types of data should we be collecting and using?). There are a wealth of interesting questions to ask and experiments to run to evaluate hypotheses that I believe this work can, in future submissions, highlight. Please see previous section for more details.

---

> ### Author Response · Authors · 2022-08-23
> **Response to Reviewer JMfs**
>
> We thank the reviewer for their detailed comments and feedback. We respond to each comment individually below.
>
> > The biggest concern with the paper is: What is the core technical contribution/novelty? The paper uses an existing data-set, together with a well-established learning procedure … to generate an encoder that is then used in a fairly standard imitation learning framework.
>
> R3M is the first paper to leverage diverse human video datasets for pre-training visual state representations for robotic manipulation. In doing so, our core contribution is an artifact – the pre-trained vision model – that can be used readily in other work. Our main contribution is not a new representation learning algorithm. We have revised the introduction of the paper to make this clearer. We believe that leveraging human videos is an essential insight and advancement over prior work in pre-training visual representations for control, which utilize static image data (RRL [1], PVR [2], CLIPort [3], EmbCLIP [4]). R3M outperforms all of these methods in a head-to-head comparison on a comprehensive simulated and real robot evaluation.
>
> [1] Shah and Kumar. RRL: Resnet as representation for Reinforcement Learning. ICML 2021.
> [2] Parisi et al. The (Un)Surprising Effectiveness of Pre-Trained Vision Models for Control. IMCL 2022.
> [3] Shridar et al. CLIPort: What and Where Pathways for Robotic Manipulation. CoRL 2021.
> [4] Khandelwal et al. Simple but Effective: CLIP Embeddings for Embodied AI. CVPR 2022.
>
> It is correct that the components of the R3M training objective are established. We emphasize the contribution of R3M is not a novel algorithm for representation learning, but a reusable artifact. While it is not the only pre-trained visual representation, our experiments suggest that it is more effective than alternatives. In fact, there are at least 4 CoRL '22 submissions that use the R3M pre-trained model, all of which are not affiliated with us. We exclude links to these submissions to preserve anonymity.
>
> > I expected to see some contribution along some axes such as: insight for general principles of what makes a data-set particularly useful for robotic application, or some novelty in training, data-augmentation, or generalization...
>
> To provide further insights, we added a new comparison that disentangles the role of the dataset and the training objective. In particular, we have trained a MoCo model on the exact same frames of the Ego4D dataset used to train our R3M model (See Table below). We then evaluated this MoCo-Ego4D model on the Franka Kitchen environment. We find that the MoCo-Ego4D model gets an average success rate of 36.0%, while R3M gets 41.2%. This suggests that while there is indeed a large benefit coming from diverse human video data compared to static ImageNet images (28% → 36%), the data is not the only source of performance improvement, and the R3M objective is particularly effective at learning a representation from such human videos, providing an additional ~5% boost in success rate.
>
> |------------|----------------|----------------|----------------|----------------|----------------|----------------|----------------|----------------|----------------|
>
> |            | Average        | Knob Turn      | Light On       | Sliding Door   | Left Door      | Microwave Open | 5 Demos        | 10 Demos       | 25 Demos       |
>
> | R3M        | **41.2 (2.6)** | **30.1 (1.6)** | **51.3 (2.7)** | **73.3 (2.1)** | **25.1 (1.6)** | **26.1 (2.0)** | **27.0 (2.1)** | **40.5 (2.4)** | **56.0 (2.6)** |
>
> | MoCo-Ego4D | 36.0 (2.4) |  28.3 (1.3) |  47.3 (2.1) | **71.6** (1.9)   | 10.7 (0.7) | 21.7 (1.4)   |  24.2 (1.9) |  36.2 (2.4) | 47.8 (2.6)  |
>
> We have added this experiment to the revision.
>
> > Beyond the above concern, the performance improvement is quite modest...We cannot conclude that R3M is a significant advance on the state-of-the-art based on these results.
>
> We humbly disagree that a performance improvement from R3M is modest.
> We would like to emphasize that the overall success rate is averaged over 12 tasks x 3 viewpoints x 3 dataset sizes = 108 different task configurations, where all tasks come from established benchmarks. Averaged over 108 configurations (and 3 seeds), a 10% absolute improvement is significant (note the standard error bars). When looking at just a single task, the R3M improvement is in some cases much larger, for example 20+% on the Franka Kitchen light task (see appendix). In fact, averaged across all tasks, the 10% improvement of R3M over the next best baseline (PVR) is comparable to the 10% improvement from PVR over learning from scratch. Finally, on our real robot evaluation, the absolute improvement of R3M over CLIP is 32%, **more than doubling the success rate of CLIP**.

---

### Author Response · Authors · 2022-08-23
**Paper Revision**

Please find attached a revised paper (changes highlighted in orange) addressing the comments of the AC and all reviewers.

---

### Meta-Review · Area_Chair_c1rS · 2022-08-06

**Recommendation:** Accept (Poster)
**Confidence:** 3

**Metareview:**

This paper proposed a pre-training method on the Ego4D dataset and used it as visual embedding for robot manipulation. This paper receives diverging reviews strong reject, weak reject, and strong accept.

Reviewers appreciate the evaluation on different sim and real-world environments. However, at the same time, all reviewers also agree that a more extended evaluation is necessary to support the paper's claims.
The major concerns are summarized below:

1. Contribution (Reviewer QF3c QzCR). What is the paper's main contribution and then to the performance improvement 1) the usage of the "Ego4D" dataset or 2) the proposed representation learning method? What are the insights a reader can learn from this paper?

2. Modest performance improvement and lack of in-depth analysis (Reviewer QF3c, JMfs). This is also related to the next point (baseline), making it hard to evaluate whether such representation can indeed advance the SOTA performance.

3. Other representation learning baselines in robotics (Reviewer QF3c,QzCR). Direct comparison to PVR or CURL (contrastive-based learning) / other imitation learning frameworks will be helpful to demonstrate whether such representation can indeed advance the SOTA performance. While CLIP is a popular visual representation for computer vision tasks, it is expected to perform poorly since it is trained on static image captioning.


During rebuttal and later discussion phase, the reviewers have a long and productive discussion.
While all reviewers agree on this paper's major strengths and weaknesses, however, different reviewer tends to weigh them differently.
- all reviewers agree this paper presents the limited novelty "uses and existing data-set with an existing architecture, with existing losses and training/algorithms"
- however, reviewers also all agree that the  "the pre-trained vision model"  does bring positive value to the robotics community.
- In terms of performance improvement, reviewers have different opinions on whether the current improvement is significant, while AC agrees with reviewer QzCR that considering the number of tasks tested, the performance improvement is convincing.

Weighting both strengths and weaknesses, AC would like to recommend accepting this paper since the pretrained weight can be a valuable resource to the community.

Author are encouraged to restructure the experiment section to include more insights and discussions. For example, "insights into why the particular model works (what is specifically useful about the data set) and what general principles we can garner/look into studying to improve our visual encoders for robotics applications (e.g., what types of data should we be collecting and using?)."


**Best Paper Nomination:**

No

---

> ### Author Response · Authors · 2022-08-23
> **Response to Meta Review**
>
>
> We thank the AC for the clear and concise summary of the reviews. We have responded to each of the reviewers individually. We highlight some key clarifications here:
>
> > (1) Contribution (Reviewer QF3c QzCR). What is the paper's main contribution and then to the performance improvement 1) the usage of the "Ego4D" dataset or 2) the proposed representation learning method?
>
> R3M is the first paper to leverage diverse human video datasets for pre-training visual state representations for robotic manipulation. In doing so, our core contribution is an artifact – the pre-trained vision model – that can be used readily in other work. Our main contribution is not a new representation learning algorithm. We have revised the introduction of the paper to make this clearer. We believe that leveraging human videos is an essential insight and advancement over prior work in pre-training visual representations for control, which utilize static image data (RRL [1], PVR [2], CLIPort [3], EmbCLIP [4]). R3M outperforms all of these methods in a head-to-head comparison on a comprehensive simulated and real robot evaluation. In fact, there are at least 4 CoRL '22 submissions that use the R3M pre-trained model, all of which are not affiliated with us. We exclude links to these submissions to preserve anonymity.
>
> [1] Shah and Kumar. RRL: Resnet as representation for Reinforcement Learning. ICML 2021.
> [2] Parisi et al. The (Un)Surprising Effectiveness of Pre-Trained Vision Models for Control. IMCL 2022.
> [3] Shridar et al. CLIPort: What and Where Pathways for Robotic Manipulation. CoRL 2021.
> [4] Khandelwal et al. Simple but Effective: CLIP Embeddings for Embodied AI. CVPR 2022.
>
> To provide further insights, we added a new comparison that disentangles the role of the dataset and the training objective. In particular, we have trained a MoCo model on the exact same frames of the Ego4D dataset used to train our R3M model (See Table below). We then evaluated this MoCo-Ego4D model on the Franka Kitchen environment. We find that the MoCo-Ego4D model gets an average success rate of 36.0%, while R3M gets 41.2%. This suggests that while there is indeed a large benefit coming from diverse human video data compared to static ImageNet images (28% → 36%), the data is not the only source of performance improvement, and the R3M objective is particularly effective at learning a representation from such human videos, providing an additional ~5% boost in success rate.
>
> |------------|----------------|----------------|----------------|----------------|----------------|----------------|----------------|----------------|----------------|
>
> |            | Average        | Knob Turn      | Light On       | Sliding Door   | Left Door      | Microwave Open | 5 Demos        | 10 Demos       | 25 Demos       |
>
> | R3M        | **41.2 (2.6)** | **30.1 (1.6)** | **51.3 (2.7)** | **73.3 (2.1)** | **25.1 (1.6)** | **26.1 (2.0)** | **27.0 (2.1)** | **40.5 (2.4)** | **56.0 (2.6)** |
>
> | MoCo-Ego4D | 36.0 (2.4) |  28.3 (1.3) |  47.3 (2.1) | **71.6** (1.9)   | 10.7 (0.7) | 21.7 (1.4)   |  24.2 (1.9) |  36.2 (2.4) | 47.8 (2.6)  |
>
> We have added this experiment to the revision.
>
> > (2) Modest performance improvement and lack of in-depth analysis (Reviewer QF3c, JMfs).
>
> We would like to emphasize that the overall success rate is averaged over 12 tasks x 3 viewpoints x 3 dataset sizes = 108 different task configurations, where all tasks come from established benchmarks. Averaged over 108 configurations (and 3 seeds), a 10% absolute improvement is significant (note the standard error bars). When looking at just a single task, the R3M improvement is in some cases much larger, for example 20+% on the Franka Kitchen light task (see appendix). In fact, averaged across all tasks, the 10% improvement of R3M over the next best baseline (PVR) is comparable to the 10% improvement from PVR over learning from scratch. Finally, on our real robot evaluation, the absolute improvement of R3M over CLIP is 32%, **more than doubling the success rate of CLIP**.
>
> > (3) Other representation learning baselines in robotics (Reviewer QF3c,QzCR). Direct comparison to PVR or CURL (contrastive-based learning) / other imitation learning frameworks will be helpful to demonstrate whether such representation can indeed advance the SOTA performance. While CLIP is a popular visual representation for computer vision tasks, it is expected to perform poorly since it is trained on static image captioning.
>
> The paper already compares directly to PVR. We followed the naming convention used in the PVR paper, referring to the method as “MoCo 345”. Critically, this is the best model trained and proposed by the PVR paper, which is the state-of-the-art in pre-trained visual representations for control. We have adjusted the naming in the revision to make this clear, and apologize for any confusion this caused.